# Assessing the Relative Impact of Diverse Stressors among Canadian Coast Guard and Conservation and Protection Officers

**DOI:** 10.3390/ijerph192416396

**Published:** 2022-12-07

**Authors:** Katie L. Andrews, Laleh Jamshidi, Jolan Nisbet, Taylor A. Teckchandani, Jill A. B. Price, Rosemary Ricciardelli, Gregory S. Anderson, R. Nicholas Carleton

**Affiliations:** 1Canadian Institute of Public Safety Research and Treatment (CIPSRT), University of Regina, Regina, SK S4S 0A2, Canada; 2Fisheries and Marine Institute, Memorial University of Newfoundland, St. John’s, NL A1C 5R3, Canada; 3Faculty of Science, Thompson Rivers University, Kamloops, BC V2C 0C8, Canada

**Keywords:** occupational stressors, public safety personnel (PSP), potentially psychologically traumatic event (PPTE), post-traumatic stress injury (PTSI)

## Abstract

Public Safety Personnel (PSP), including members of the Canadian Coast Guard (CCG) and Conservation and Protection (C&P) officers, are regularly exposed to potentially psychologically traumatic events (PPTEs) and other occupational stressors (organizational and operational stressors). The current study quantified occupational stressors among CCG and C&P and assessed relationships with PPTEs and mental health disorders. Participants (*n* = 341; 58.4% male) completed an online survey assessing self-reported occupational stressors, PPTEs, and mental health disorder symptoms. CCG and C&P Officers reported significantly lower mean overall and item-level organizational and operational stress scores compared to other Canadian PSP. Mean operational stress scores were statistically significantly associated with increased odds of screening positive for all mental disorders and organizational stress scores were statistically significantly associated with increased odds of screening positive for all mental disorders except social anxiety disorder. Participants reported several item-level occupational stressors associated with screening positive for posttraumatic stress disorder, general anxiety disorder, major depressive disorder, social anxiety disorder, panic disorder, and alcohol use disorder, even after accounting for diverse PPTE exposures. Exposure to PPTEs may be a regular part of employment for CCG and C&P PSP; however, bureaucratic red tape, staff shortages, excessive administrative duties, physical conditioning, healthy eating, and fatigue are occupational stressors that appear significantly related to mental health. Ongoing mental health efforts are needed to mitigate and manage the impact of occupational stressors among CCG and C&P.

## 1. Introduction

Public safety personnel (PSP) include, but are not limited to, border services officers, correctional workers, firefighters (career and volunteer), Indigenous emergency management, operational and intelligence personnel, paramedics, policing (municipal and provincial), public safety communication, Royal Canadian Mounted Police, and search and rescue personnel [1]. At least two operating agencies within the Department of Fisheries and Oceans Canada (DFO) also include PSP; specifically, the Canadian Coast Guard (CCG) and Conservation and Protection Services (C&P). The CCG helps to ensure Canada’s sovereignty and security by maintaining a presence in Canadian waters, conducts search and rescue operations, and provides marine assistance across Canada [2]. C&P Officers have duty-specific responsibilities related to law enforcement and the protection of species at risk, fish habitat and oceans, and carry out a wide range of duties, both on land and at sea, overtly and covertly, and in remote locations [3].

As a function of their occupations, PSP are frequently exposed to a wide variety of workplace stressors. Research examining workplace stressors has focused on exposures to potentially psychologically traumatic events (PPTEs) [4,5,6]. PSP report much higher frequencies of PPTE exposures than the general population [4]. Canadian PSP have reported a lifetime average of exposure to 11 different PPTE types [4]. CCG and C&P previously reported exposure to an average of eight different PPTE types, with each type being experienced 10 or more times by up to 78.9% of respondents [7]. Research including PSP has demonstrated that PPTE exposure is associated with increased risk for the development of mental health disorders (e.g., post-traumatic stress disorder [PTSD], major depressive disorder [MDD], panic disorder [PD], generalized anxiety disorder [GAD], social anxiety disorder [SAD], alcohol use disorder [AUD]) [8]. Approximately 42.0% of CCG and C&P respondents [9] and 44.5% of other Canadian PSP [8] screened positive for one or more mental health disorder. The apparent mental health challenges reported among PSP have led to an increased interest in further examining the associations between workplace stressors and mental health outcomes among all PSP groups, including CCG and C&P PSP [10].

PPTE exposures among PSP appear to be unavoidable workplace stressors that can negatively impact mental health outcomes. Previous research has suggested that PSP report experiencing a wide range of other occupational stressors categorized into two construct groups: organizational stressors and operational stressors [11]. Organizational stressors are frequently defined as the stressors associated with job context or setting, such as staff shortages, a lack of appropriate resources, inconsistent leadership styles, unequal sharing of work responsibilities, and differential treatment of employees by leadership [11,12,13]. Operational stressors typically refer to the stressors directly tied to work content and duties, such as fatigue from shift work and overtime, risk of being injured on the job, social and personal life limitations, management of behaviors related to health and fitness, and the inescapability of work [11,12,13]. Organizational and operational stressors have been assessed in a variety of PSP occupations (e.g., police, firefighters, and correctional officers) [11,14,15,16,17,18,19] including a diverse national sample of Canadian PSP [20].

Research examining occupational stressors among PSP and associations with mental health challenges remains limited. Research regarding the unique impact of PPTEs, organizational and operational stressors, and whether occupational stressors and PPTEs interact to adversely influence PSP mental health is also lacking. The impact of PPTEs relative to organizational and operational stressors is worth investigating due to the nature of PSP work which involves necessary, unavoidable, and repeated PPTE exposures. Effectively managing other occupational stressors may help to mitigate the risk of mental health challenges among PSP. Therefore, researchers, organization leadership, and policy makers may want to shift their focus to the potential benefits of modifying specific organizational and operational stressors to protect PSP mental health. Indeed, operational stressors may be inherent to the job, but employers can always work to remedy organizational stressors. In a sample of Canadian PSP, participants reported substantial difficulties with most organizational and operational stressors [20]. The same results indicated that occupational stressors were uniquely associated with development of positive screens for all measured mental health disorders, including PTSD. Organizational and operational stressors remained statistically significantly associated with positive screens for all measured mental health disorders after controlling for sociodemographic factors and PPTE exposures.

Despite the recognition that the CCG and C&P include PSP, researchers have yet to assess occupational stressors among these groups. There is currently no known published research that has examined occupational stressors among CCG and C&P PSP and assessed for associations with PPTEs and mental health disorders. Demonstrating associations between occupational stressors and mental health disorders, after controlling for the associations of PPTEs, would provide support for efforts to modify specific organizational and operational stressors. CCG and C&P leadership, managers, and policy makers can use evidence regarding specific occupational stressors to design and implement training, interventions, and resources to mitigate and manage mental health challenges of their personnel.

The objectives of the current study were to assess for: (1) sociodemographic differences in mean overall organizational and operational stress scores among CCG and C&P officers; (2) differences in overall and item-level organizational and operational stressors between CCG and C&P officers and between the total sample and a diverse sample of previously surveyed Canadian PSP (i.e., municipal/provincial police, firefighters, paramedics, Royal Canadian Mounted Police, correctional workers, dispatchers); (3) associations between positive screens on several mental health outcomes (i.e., PTSD, MDD, GAD, SAD, PD, and AUD) and the overall and item-level scores for operational and organizational stressors; (4) unique associations between overall and item-level operational and organizational stress scores and a positive screen on several mental health outcomes, after controlling for PPTE exposure types; and (5) the extent to which the interaction between PPTE exposure and occupational stress predicted a positive screen for mental health.

Variation in overall and item-level organizational and operational stressors was expected, but there were no specific directional hypotheses about which stressors would be higher than others. Statistically significant and positive associations were also expected between overall and item-level organizational stressors and screening positive for a mental health disorder. The associations were expected to remain significant after controlling for PPTE exposures. A statistically significant interaction of stressors (e.g., high PPTE exposures and occupational stressors) was expected alongside positive screens for mental health disorders.

## 2. Materials and Methods

### 2.1. Procedure

Data were collected using a web-based self-report survey available in both English and French. The study was approved by the University of Regina Institutional Research Ethics Board (REB# 2021-003). The survey was based on a set of validated measures used in a previous study of PSP [4,5,6,8,20,21], but collaboratively redesigned by the research team and the CCG and DFO team to ensure relevant variables were included. The survey was promoted and distributed by the CCG and DFO to member unions via emails, social media posts, and a video encouraging participation. The survey was available from 1 February 2021 to 31 January 2022. At the start of the survey, participants selected their preferred language (i.e., English or French) in which to complete the survey and were presented with study information and informed consent. Participation was anonymous and voluntary, and each respondent was provided with a randomly generated unique code which allowed for repeated survey access to complete the survey over multiple sessions. The current study focused specifically on the self-reported organizational and operational stressors, PPTE exposures, and positive screens for mental health disorders based on several well-established measures assessing mental disorder symptoms.

### 2.2. Data and Sample

Participants were CCG and C&P PSP (*n* = 341) (70.4% CCG members and 29.3% C&P members). Responses from 561 CCG and C&P PSP were initially collected; however, only data from respondents who completed at least 30% of the survey were retained. For the current study, data from respondents who completed the occupational stress questionnaires were included in the current analyses and results. The final sample included a total of 341 respondents. Participants were mainly white (i.e., Caucasian) (87.7%), male (58.4%), identifying as men (57.5%), aged 30–39 (28.4%) and 40–49 (27.6%) years old (see Table 1). Participants were mostly married or in common law relationships (i.e., living with a person in a conjugal relationship for 12 continuous months) (68.0%), with a college (39.9%) or university (34.9%) degree, residing in British Columbia (56.6%), with no previous work experience as PSP or in the Canadian Armed Forces (71.0%).

### 2.3. Self-Report Measures

Occupational Stressors. Occupational stressors were assessed with the 20-item Organizational Police Stress Questionnaire (PSQ-Org) and the 20-item Operational Police Stress Questionnaire (PSQ-Op) [11]. The PSQ-Org assessed stressors associated with the organization and culture within which the job is performed, including the impact of work on family and social life (e.g., fatigue, occupation-related health issues, not enough time to spend with friends and family). The PSQ-Op assessed stressors associated with the job (e.g., dealing with co-workers, staff shortages, inconsistent leadership). Despite the titles of the scales, the items are not specific to policing, such that each item can apply to other PSP professionals; indeed, the scales have been used successfully with a wide range of PSP [20]. Each item on both the PSQ-Org and PSQ-Op is rated on a 7-point Likert scale ranging from 1 (no stress at all) to 7 (a lot of stress). The overall mean scores on the PSQ-Org and PSQ-Op were computed separately by summing responses across all the items and dividing by 20, as per the measure-specific instructions [11,15,21]. Individual means were also computed for each PSQ-Orq and PSQ-Op item.

Potentially Psychologically Traumatic Events (PPTEs). The survey included the Life Events Checklist for the Diagnostic and Statistical Manual of Mental Disorders 5-Extended (DSM-5) (LEC-5) [22]. The LEC-5 does not include the unexpected death of a loved one, an adverse event that no longer meets criteria for PTSD in the DSM-5 [23]. Participants reported on the PPTE exposure modality (e.g., indirectly or directly) and all their reported experiences were treated as exposures for the current article: (a) it happened to them personally; (b) they witnessed it happen to someone else; (c) they learned about it happening to a close family member or close friend; and/or (d) they were exposed to it as part of their job. The total number of different PPTE exposure types was quantified by summing exposure frequencies across the 17 items. The LEC-5 was modified to ask participants to indicate the number of exposures to each PPTE type they experienced. Participants who reported exposure to more than one PPTE type were asked to select the worst PPTE or the PPTE currently causing them the most distress, as well as the number of exposures to that PPTE type, and the length of time since the first and the last exposure (i.e., most recent).

Mental Health Disorder Symptoms. Mental health disorder symptoms were assessed by self-report using the PTSD Checklist for DSM-5 (PCL-5) [24,25]; the 9-item Patient Health Questionnaire (PHQ-9) [26] indexing MDD symptoms; the Panic Disorder Symptoms Severity scale, Self-Report (PDSS-SR) [27] indexing panic disorder (PD) symptoms; the 7-item Generalized Anxiety Disorder scale (GAD-7) [28] indexing GAD symptoms; the Social Interaction Phobia Scale (SIPS) [29] indexing SAD symptoms; and the Alcohol Use Disorders Identification Test (AUDIT) [30] indexing alcohol use disorder (AUD) symptoms. Participants reported their behaviors over the last year for the AUDIT, the past month for the PCL-5, the past 14 days for the PHQ-9 and GAD-7, and the past 7 days for the PDSS-SR. There is no specific time window used for SIPS. For the PCL-5, a positive screen required participants to report exposure to at least one LEC-5 item, meet minimum DSM-5 [23] criteria for each PTSD symptom cluster subscale (e.g., intrusions, avoidance, negative alterations in cognitions and mood, and alterations in arousal and reactivity), and exceed the clinical cut-off of >32 [24]. A positive screen required the PHQ-9 total score to be >9 [8], the PDSS-SR total score to be >7 [27], the GAD total score to be >9, [28] the SIPS total score to be >20 [29], and the AUDIT total score to be >15 [31].

### 2.4. Statistical Analyses

To examine the first research objective, means and standard deviations for overall organizational and operational stress scores for the total sample were computed across sociodemographic categories. Independent samples *t*-tests and analysis of variance tests (ANOVAs) were conducted to assess statistically significant differences in overall mean organizational and operational stress scores across sociodemographic categories. Holm–Bonferroni adjustments were applied to alpha levels in post hoc tests to reduce familywise error rate. To examine the second research objective, one-sample t-tests were conducted to compare means of overall and item-level organizational and operational stressors between the CCG and C&P samples and between the total sample and previously surveyed Canadian PSP [20].

To examine the third and fourth research objectives, multivariate logistic regression models were conducted to examine associations between organizational and operational stressors overall and item-level means and positive screens for mental disorders. All logistic regression models were adjusted for sociodemographic covariates (i.e., sex, gender, age, education, ethnicity marital status, province of residence, previous work experience), the total number of PPTE types (range 0 to 16), and job category (i.e., CCG and C&P).

To examine the fifth research objective, a series of nested multivariate logistic regression analyses were conducted to assess the independent and interactive effects of mean organizational and operational stress scores and the total number of PPTE exposures on each specific positive mental disorder screens. The nested multivariate logistic regression models were adjusted for sociodemographic covariates and job category. Calculated adjusted odds ratios (AORs) described the associations between positive mental disorder screens and the total number of PPTE exposures (model 1), the mean organizational stress scores (model 2), and the mean operational stress scores (model 3). Models 1, 2, and 3 examined the individual associations between these predictors (i.e., the total number of PPTE exposures, the mean organizational stress scores, and the mean operational stress scores) and positive screens of mental disorders. Model 4 included all predictors (i.e., the total number of PPTE exposures, the mean organizational stress scores, and the mean operational stress scores) and examined the independent associations for each predictor after controlling for the other two predictors and mental disorder positive screens. Models 5 and 6 added the interactive effects of PPTE and each occupational stress measure (i.e., the mean organizational stress scores and the mean operational stress scores) to Model 4, adjusting for covariates and the other occupational stress measure.

## 3. Results

### 3.1. Assessing Organizational and Operational Stress across Sociodemographic Categories

The following results address the first research objective. Mean organizational and operational stress scores across different sociodemographic categories are presented in Table 2. Individuals aged 40 to 49 years old reported statistically significantly higher overall mean organizational stress scores than individuals between the age of 19 and 29 years old (*p* < 0.01). Statistically significant effects were observed for ethnicity and mean operational stress scores; however, follow-up multiple pairwise comparisons were not significant due to application of the Holm–Bonferroni adjustment to alpha levels for post hoc test to control for any potential familywise errors.

### 3.2. Assessing Organizational and Operational Stressors across Job Categories

The following results address the second research objective. Overall and item-level mean organizational and operational stress scores for the total, CCG, and C&P samples are provided in Table 3. For the total sample, the specific organizational stressors with the highest mean levels of stress were bureaucratic red tape (3.14 ± 2.01), staff shortages (3.10 ± 2.17), excessive administrative duties (3.02 ± 2.02), dealing with co-workers (2.81 ± 1.87), and constant change in policy/legislation (2.73 ± 2.00). For the total sample, the operational stressors with the highest mean levels of stress were finding time to stay in good physical condition (2.64 ± 1.97), fatigue (e.g., shift work, over-time) (2.23 ± 2.08), paperwork (2.12 ± 1.79), not enough time available to spend with friends and family (2.10 ± 1.97) and eating healthy at work (1.88 ± 1.77).

There were statistically significant differences between CCG and C&P on the overall scores of organizational and operational stress measures and some individual items (see Table 3). C&P Officers reported statistically significantly higher mean organizational stress scores (*p* < 0.05, *d* = 0.278) and mean operational stress scores (*p* < 0.01, *d* = 0.318) than CCG participants. C&P Officers also reported statistically significantly higher mean scores on some item-level organizational stressors (i.e., excessive administrative duties, constant change in policy/legislation, bureaucratic red tape, too much computer work, lack of resources, and dealing with the court system; all *p*s < 0.05) and some item-level operational stressors (i.e., work-related activities on days off [e.g., court, community events], paperwork, upholding a “higher image” in public, negative comments from the public, limitations to your social life [e.g., who your friends are, where you socialize], and friends/family feel the effects of the stigma associated with your job; all *p*s < 0.05) compared to CCG participants. The total sample reported statistically significantly lower mean stress scores for overall and all item-level organizational and operational stressors compared to a previously published sample of diverse Canadian PSP [18] (all *p*s < 0.001).

### 3.3. Associations between Operational and Organizational Stressors and Mental Health

The following results address the third research objective. Associations between overall and item-level organizational and operational stress scores and positive screens for mental health disorders are presented in Table 4. The results are provided as odds ratios adjusted for sociodemographic covariates, the total number of PPTE exposures, and occupation category. The overall mean scores for organizational (adjusted odds ratios (AORs) ranged from 1.39 to 2.21) and operational stress (AORs ranged from 1.65 to 2.19) were both statistically significantly associated with increased odds of screening positive for all mental disorders, except for the association between overall organizational stress score and SAD. Most organizational (i.e., 14 items; AORs ranged from 1.21 to 1.57) and all operational stressors (i.e., 20 items; AORs ranged from 1.20 to 1.57) were statistically significantly associated with increased odds of screening positive for PTSD. Several organizational (i.e., 8 items; AORs ranged from 1.16 to 1.34) and operational stressors (i.e., 15 items; AORs ranged from 1.16 to 1.41) were statistically significantly associated with increased odds of screening positive for MDD. Several organizational (i.e., 7 items; AORs ranged from 1.19 to 1.39) and operational stressors (i.e., 14 items; AORs ranged from 1.19 to 1.43) were statistically significantly associated with increased odds of screening positive for GAD. Some organizational (i.e., 3 items; AORs ranged from 1.26 to 1.27) and operational stressors (i.e., 8 items; AORs ranged from 1.20 to 1.49) were statistically significantly associated with increased odds of screening positive for SAD. Several organizational (i.e., 8 items; AORs ranged from 1.28 to 1.54) and operational stressors (i.e., 12 items; AORs ranged from 1.26 to 1.79) were statistically significantly associated with increased odds of screening positive for PD. Several organizational (i.e., 13 of 20 items; AORs ranged from 1.24 to 1.56) and operational stressors (i.e., 10 of 20 items; AORs ranged from 1.26 to 1.66) were statistically significantly associated with increased odds of screening positive for AUD.

### 3.4. Unique Associations between Organizational Stress, Operational Stress, and Mental Health

The following results address the fourth and fifth research objectives. Independent and interactive effects of mean organizational and operational stress overall scores and the number of PPTE exposures on positive screens for mental disordersare presented in Table 5. The results are presented as odds ratios, adjusted for sociodemographic covariates. Models 1, 2, and 3 entered, respectively, the total number of PPTE exposures, the mean overall organizational stress scores, and the mean operational stress scores independently into the models to assess their individual baseline effects. Model 1 indicated that statistically significant associations were observed between the total number of PPTE exposures and PTSD (AOR, 1.13; 95% confidence interval (CI), 1.03–1.25) and PD (AOR, 1.14; 95% CI, 1.00–1.29). Model 2 indicated statistically significant associations between mean organizational stress scores and increased odds of screening positive for all mental disorders except SAD (AORs ranged from 1.42 to 1.92). Model 3 indicated statistically significant associations between the mean operational stress scores and screening positive for all mental disorders (AORs ranged from 1.60 to 2.37). Model 4 examined unique associations between the total number of PPTE exposures, the mean organizational stress scores, and the mean operational stress scores as predictors, and each positive mental disorder screens as dependent variables. When all predictors were entered into Model 4 simultaneously, the total number of PPTE exposures and the mean organizational stress scores were statistically significantly associated with AUD (AORs = 0.87 and 2.31, respectively) and the mean operational stress scores remained independently associated with increased odds of screening positive for all mental disorders except for AUD (AORs ranged from 1.97 to 2.34). Models 5 and 6 examined the interactive effects of organizational and operational stress and the number of PPTE exposures on screening positive for mental disorders. The interactive effects were not statistically significantly associated with screening positive for any mental health disorders.

## 4. Discussion

The current study assessed differences in overall and item-level organizational and operational stressors among CCG and C&P officers and assessed for independent and unique interactive associations between operational and organizational stress scores, PPTE exposures, and screening positive for mental health disorders. The results provide evidence that PPTEs are only one factor associated with mental health challenges among CCG and C&P PSP. Operational and organizational stressors relevant to job duties for PSP were both statistically significantly associated with screening positive for mental health disorders, with moderate to large effect sizes, while statistically controlling for the influence of PPTEs. The current results provide evidence that CCG and C&P leaders and managers need to understand the role workplace stressors play on the mental health of their personnel. The current results suggest that organizational and operational stressors, not just PPTEs, have a large impact on PSP mental health. Effectively managing the occupational stressors identified as most impactful to CCG and C&P officers could be, a least in part, one solution to reduce the risk of mental health challenges among CCG and C&P officers.

Participating CCG and C&P PSP reported difficulties with various types of PPTE exposures (i.e., mean score of 8.72 out of 16) [7]; however, participants also reported substantial difficulties with several organizational stressors (i.e., mean scores from 2.42 to 3.14 out of 7), such as bureaucratic red tape, staff shortages, excessive administrative duties, dealing with co-workers, and constant changes in policy/legislation. Bureaucratic red tape, staff shortages, and dealing with co-workers were also among the highest organizational stressors reported by a diverse sample of other Canadian PSP [20]. Substantial difficulties were also reported with several operational stressors (i.e., mean scores from 2.10 to 2.64 out 7), such as finding time to stay in good physical condition, fatigue, paperwork, not enough time available to spend with friends and family, and eating healthy at work. Finding time to stay in good physical condition, fatigue associated with shift work and overtime, and not enough time available to spend with family and friends were also among the highest operational stressors reported by other Canadian PSP [18].

The highest operational stressors reported by CCG and C&P appear related to the impacts of work on their health, fitness, and social support. Several of these stressors, including fatigue, shift work, working alone, and staff shortages were identified as potentially problematic in previous research conducted in other occupations [32,33,34,35] and were associated with increased odds of screening positive for mental disorders among other Canadian PSP [20]. The current results identify the specific organizational and operational stressors most impacting CCG and C&P officers that need to be addressed by organizational leaders and managers in order to mitigate the mental health challenges faced by their members. Solutions could be implemented to help members manage their health and fitness (i.e., access to gyms and time to use them) and manage their fatigue and access their social supports (i.e., additional breaks and time off). Further research is needed to assess what solutions CCG and C&P officers would find most helpful in reducing the impact of occupational stressors.

There were statistically significant differences between the CCG and C&P participants’ levels of stress for both organizational and operational stressors. C&P Officers reported statistically significantly higher mean organizational and mean operational stress scores than CCG personnel. Several statistically significant differences were identified among the item-level stressors that may provide insight for leaders of each organization. For organizational stressors, higher levels of stress were reported by C&P Officers for excessive administrative duties, bureaucratic red tape, too much computer work, lack of resources, and dealing with the court system. C&P Officers also reported higher levels of stress for operational stressors including paperwork, negative comments from the public, limitations to your social life, upholding a “higher image” in public, friends and family feel the effects of the stigma associated with your job, and work-related activities on days off (e.g., court, community events).

Differences between CCG and C&P Officers levels of stress reported for specific organizational and operational stressors may be due to differences in occupational duties. C&P Officers may engage in duties specific to law enforcement and encounter confrontational members of the public in remote locations, with little to no back-up or assistance [36]. There is also a small number of C&P Officers (~600) spread across Canada [3], which increases the likelihood of working independently and having less access to supports [37,38]. The observed differences in organizational and operational stressors offer specific directions for leaders looking to make changes to minimize mental health challenges experienced by PSP. Each stressor could be addressed and, where possible, CCG and C&P organization leaders could tailor the resources or solutions implemented to reduce the impact of the stressor on personnel. A particular focus on organizational stressors is necessary as each is a direct result of employer practices and thus amendable with creative solutions to such stressors, which contrasts with most operational stressors that are inherent to the job and often unavoidable (i.e., responding to calls for service or adverse events).

There were also significant differences between the CCG and C&P participants and other Canadian PSP [20]. The total sample reported significantly lower mean organizational and operational stress scores, as well as significantly lower stress scores for all item-level organizational and operational stressors compared to other Canadian PSP [20]. Differences in levels of occupational stress may be due to diverse factors related to differences across the PSP organizations. The differences may be associated with leadership, management structures, workload expectations, environmental variables, or other currently unidentified factors. Nevertheless, the current results support the contention that CCG and C&P Officers are exposed to diverse occupational stressors that are potentially problematic for PSP and play a role in their mental health challenges.

Among CCG and C&P Officers, both organizational and operational stressors were associated with adverse outcomes. Mean operational stress scores were statistically significantly associated with screening positive for all mental health disorders included in the analyses. Mean organizational stress scores were associated with screening positive for all mental disorders analyzed, except SAD. The results suggest that organizational and operational stressors may be of particular interest to managers and organizational leaders as targets for interventions to mitigate and manage mental health challenges. Several item-level organizational and operational stressors were statistically significantly associated with increased odds of screening positive for mental disorders. All item level operational stressors and many organizational stressors (14 out of 20) were statistically significantly associated with increased odds of screening positive for PTSD. Few item-level organizational stressors (3 out of 20) and only some operational stressors (8 out of 20) were associated with SAD. CCG and C&P PSP have previously reported a statistically significantly higher prevalence of SAD (21.4%) than other Canadian PSP [9]. The current results suggest that among CCG and C&P occupational stressors are associated with increased odds of screening positive for mental health disorders such as PTSD; however, other factors unique to CCG and C&P duties could be associated with increased odds of screening positive for SAD, such as social isolation or perceived social support [20,39]. Lower perceived social support has been reported to be associated with increased odds of SAD and other anxiety disorders [40]. CCG and C&P organizational leaders could use the current results to tailor interventions and resources to address the specific organizational and operational stressors impacting their members to reduce their risk of screening positive for mental health disorders. Future research is needed to assess the effectiveness of any implemented solutions.

Mean organizational and operational stress scores were associated with the highest odds of screening positive across all mental health disorders; however, there were some notable stressors associated with the highest odds of screening positive for specific mental disorders. The organizational stressor with the highest odds of screening positive for all mental health disorders except AUD was believing if you are sick or injured, your co-workers seem to look down on you. Due to stigma, CCG and C&P members may avoid disclosing mental health challenges, illnesses, or injuries for fear of being perceived as unfit for duty, which could further reduce their supports and treatment seeking behaviors [41,42]. Indeed, the ability to go on ship is contingent on being mentally well, thus disclosing mental health challenges may, in reality, prevent continued operational deployment. Stigma against co-workers with mental health disorders has been reported to be associated with low levels of intentions to seek mental health services among Canadian PSP [43], which may further exacerbate the mental health challenges faced by CCG and C&P. Organization leaders may include mental health training and resources to increase members’ mental health knowledge to reduce levels of stigma [44] and negative attitudes, and in turn mitigate levels of stress of members.

The operational stressors with the highest odds of screening positive for all mental disorders except AUD were managing your social life outside of work and making friends outside the job. These specific stressors appear mainly related to social supports, which is a potential protective factor for mental health among PSP [38,45,46]. Previous research including other PSP occupations evidenced greater social support as being associated with decreased likelihood of screening positive for PTSD and MDD [46]. The current results suggest that CCG and C&P PSP may lack social support or perceived social support and highlight the potential role of social support as a target for organization leaders when designing and implementing interventions to mitigate mental health challenges among PSP. Overall, the highlighted specific organizational and operational stressors associated with the highest odds of screening positive for mental disorders can inform organization leadership about ways to tailor their efforts to address the specific needs of their members.

The total number of PPTE exposure types was independently associated with increased odds of screening positive for PTSD. This is consistent with previous research including other Canadian PSP [20]. Mean operational stress scores were associated with increased odds of screening positive for all mental disorders, while mean organizational stress scores were associated with increased odds of screening positive for all mental disorders except SAD. When assessed together, the total number of PPTE exposure types and mean organizational stress scores remained associated with AUD, whereas mean operational stress score remained independently associated with increased odds of screening positive for all mental disorders. This is consistent with previous research including other Canadian PSP [20]; however, in the previous study both organizational and operational stress remained significantly independently associated with screening positive for all mental health disorders. The current results suggest operational stressors may play a relatively larger role on the mental health of CCG and C&P PSP than organizational stressors. The interactive associations were not significant, suggesting against a moderating effect, such that both PPTE exposure types, and both organizational and operational stressors, appear independently associated with mental health challenges. The results suggest that if PPTE exposures are unavoidable, leadership may still have an opportunity to mitigate and manage mental health challenges by implementing resources to reduce occupational stressors.

### Strengths and Limitations

The use of data provided by a national and diverse sample of CCG and C&P personnel is an important strength of the current study; however, there are several limitations that provide directions for future research. First, the current sample may not be entirely representative of the entire CCG and C&P working across Canada. The CCG and C&P includes approximately 6700 members (i.e., CCG 6100, C&P 600) [2,3]. The current sample reflects approximately 5.08% of CCG and C&P members and includes larger proportions of CCG members (70.4%) than C&P members (29.3%), relatively larger proportions of members from British Columbia, and smaller proportions of members from Quebec, Nova Scotia, Newfoundland, and Labrador.

Second, participation in the current study was anonymous, voluntary, and self-selected. The recruitment materials described the study as focusing on occupational stressors, mental health disorders, and PPTEs, which may have attracted participants who were experiencing high levels of stress. The collection method using an online survey may have also impacted the number of participants. Many CCG and C&P members do not have easy access to computers or the internet as they serve on ships, stations, or in the field, and are often away for long periods of time. Participants were able to begin, leave, and return to the survey at their leisure, to ease survey response burden; as such, there is no way to know the average length of survey completion time or to understand why some participants did not complete the entire survey or only completed specific questionnaires.

Third, the screening measures for mental health disorders used in the current study are valid and reliable for use in clinical settings; nevertheless, diagnoses can only be made using clinical interviews with supporting collateral information. Further, only a relatively small number of potential mental health disorders were screened for in the current study. Future research should consider including clinical interviews to provide diagnoses and examine additional mental health disorders.

Despite the limitations, the survey sociodemographics indicate that the sample was generally proportionally consistent with the age and sex of CCG and C&P personnel. The current study provides the first known national information on occupational stressors and assessed associations with mental health disorder positive screenings and PPTE exposures among CCG and C&P members. The selected measures allow for comparisons with other large occupational studies designed to estimate occupational stressors in specific occupational samples such as PSP [20]. The current results provide potentially important information to support researchers and organization leadership interested in possible ways to mitigate and manage occupational stressors and reduce PTSI among PSP.

## 5. Conclusions

The current results provide information on occupational stressors among CCG and C&P PSP. The current results suggest that organizational and operational stressors, not just PPTEs, have a large impact on PSP mental health. Overall, participants reported significant occupational stress associated with organizational and operational stressors. Organizational stress scores were statistically significantly associated with screening positive for all mental disorders except SAD. Operational stress scores were statistically significantly associated with screening positive for all mental disorders. All item-level occupational stressors were associated with screening positive for at least one mental disorder. The most item-level stressors were associated with PTSD, while the lowest number with SAD. The current results provide evidence that CCG and C&P leaders and managers need to understand the role workplace stressors play on the mental health of their personnel and suggest that a successful action plan to address mental health challenges among CCG and C&P should include changes to reduce organizational and operational stressors. Focusing on reducing the impact of specific occupational stressors, reducing stigma, and strengthening social support appear to be promising opportunities for organization leadership who want to implement solutions to protect the mental health of personnel. Future research is needed to assess what solutions CCG and C&P officers would find most helpful in reducing the impact of occupational stressors and to assess the effectiveness of implemented solutions.

## Figures and Tables

**Table 1 ijerph-19-16396-t001:** Participant Sociodemographics Information and Distribution (*n* = 341).

Categories	% (*n*)
Gender	
Man	57.5 (196)
Woman	40.8 (139)
Non-Binary	^
Two-Spirits	^
Sex	
Male	58.4 (199)
Female	41.1 (140)
Age	
19–29	12.9 (44)
30–39	28.4 (97)
40–49	27.6 (94)
50–59	24.3 (83)
60+	5.3 (18)
Education	
High School or Less	8.5 (29)
College Program (e.g., Trade School; 2-Year College Diploma)	39.9 (136)
Coast Guard College: Graduated Fleet	9.7 (33)
Coast Guard College: MCTS Officer Training	2.3 (8)
University Degree (4-year College or Higher)	34.9 (119)
Ethnicity	
Asian	2.3 (8)
Black	^
Hispanic	^
Indigenous (i.e., First Nations, Inuit, Métis)	3.5 (12)
South Asian	^
White	87.7 (299)
Prefer not to answer	1.5 (5)
Other	3.8 (13)
Marital Status	
Single	22.6 (77)
Married/Common Law	68.0 (232)
Separated/Divorced	7.3 (25)
Widowed	^
Province of residence	
British Columbia	56.6 (193)
New Brunswick	1.5 (5)
Newfoundland and Labrador	7.6 (26)
Northern Territories (YK, NWT, NVT)	^
Nova Scotia	10.3 (35)
Ontario	11.4 (39)
Québec	12.0 (41)
Previous Work Experience	
Neither	71.0 (242)
Public Safety Only	18.2 (62)
CAF Only	8.5 (29)
CAF and Public Safety	2.3 (8)
Job Category	
CCG	70.4 (240)
C&P	29.3 (100)
Not specified	^
Total Sample	100(341)

Note. Total percentages may not sum to 100 and *n*s may not sum to 341 due to non-response or responding “other”. ^: Sample size between 1 and 4, so data not presented. CAF = Canadian Armed Forces; CCG = Canadian Coast Guard; C&P = Conservation and Protection; MCTS = Marine Communications and Traffic Services; NWT = Northwest Territories; NVT = Nunavut; YK = Yukon.

**Table 2 ijerph-19-16396-t002:** Occupational Stressors Levels across Participant Demographics Categories.

	Organizational Stressors		Operational Stressors	
	*n*	*Mean (SD)*	*n*	*Mean (SD)*
**Total Sample**	341	2.11 (1.25)	340	1.49 (1.09)
Gender				
Man	196	2.11 (1.22)	196	1.52 (1.10)
Woman	139	2.09 (1.27)	138	1.41 (1.06)
Non-Binary	^	^	^	^
Two-Spirits	-	-	-	-
Test Statistic ^1^	^	*F* (3337) = 2.26	^	*F* (3336) = 2.23
Effect Size ($\eta_{p}^{2}$)	-	0.020	-	0.019
Sex				
Male	199	2.12 (1.22)	199	1.54 (1.10)
Female	140	2.08 (1.28)	139	1.41 (1.06)
Test Statistic ^1^	-	*t* (337) = 0.35	-	*t* (336) = 1.08
Effect Size (Cohen’s d)	-	0.039	-	0.119
Age				
19–29	44	1.48 (0.99) ^b^	44	1.40 (0.96)
30–39	97	2.09 (1.21) ^a,b^	96	1.57 (1.13)
40–49	94	2.39 (1.31)^a^	94	1.59 (1.13)
50–59	83	2.22 (1.25)^a,b^	83	1.49 (1.03)
60+	18	1.88 (1.32) ^a,b^	18	1.04 (1.25)
Test Statistic ^1^	*-*	*F* (4331) = 4.48 **	-	*F* (4330) = 1.17
Effect Size ($\eta_{p}^{2}$)	-	0.051	-	0.014
Education				
High School or Less	29	1.58 (1.36)	29	1.17 (1.23)
College Program (e.g., Trade School; 2-Year College Diploma)	136	2.27 (1.29)	135	1.54 (1.16)
Coast Guard College: Graduated Fleet	33	1.99 (1.06)	33	1.49 (0.64)
Coast Guard College: MCTS Officer Training	8	1.69 (1.07)	8	1.73 (0.99)
University Degree (4-year College or Higher)	119	2.10 (1.19)	119	1.45 (1.06)
Test Statistic ^1^	-	*F* (4320) = 2.21	-	*F* (4319) = 0.83
Effect Size ($\eta_{p}^{2}$)	-	0.027	-	0.010
Ethnicity				
Asian	8	2.23 (0.80)	7	1.61 (0.93)
Black	^	^	^	^
Hispanic	^	^	^	^
Indigenous (e.g., First Nations, Inuit, Métis)	12	1.92 (1.35)	12	0.80 (0.75)
South Asian	^	^	^	^
White	299	2.09 (1.23)	299	1.48 (1.06)
Prefer not to answer	5	3.78 (0.81)	5	3.31 (1.05)
Other	13	2.13 (1.57)	13	1.58 (1.36)
Test Statistic ^1^	-	*F* (7333) = 1.60	-	*F* (7332) = 2.99 **
Effect Size ($\eta_{p}^{2}$)	-	0.033	-	0.059
Marital Status				
Single	77	1.93 (1.21)	76	1.53 (1.05)
Married/Common Law	232	2.13 (1.26)	232	1.46 (1.09)
Separated/Divorced	25	2.45 (1.23)	25	1.56 (1.17)
Widowed	^	^	^	^
Test Statistic ^1^	-	*F* (3332) = 1.18	-	*F* (3331) = 0.13
Effect Size ($\eta_{p}^{2}$)	-	0.011	-	0.001
Province of residence				
British Columbia	193	2.25 (1.24)	192	1.59 (1.12)
New Brunswick	5	1.98 (1.07)	5	1.61 (0.90)
Newfoundland and Labrador	26	1.54 (1.19)	26	1.40 (1.09)
Northern Territories (YK, NWT, NVT)	^	^	^	^
Nova Scotia	35	2.19 (1.39)	35	1.48 (1.17)
Ontario	39	1.97 (1.10)	39	1.14 (0.85)
Québec	41	1.91 (1.30)	41	1.41 (1.05)
Test Statistic ^1^	-	*F* (6333) = 1.61	-	*F* (6332) = 1.20
Effect Size ($\eta_{p}^{2}$)	-	0.028	-	0.021
Previous Work Experience				
Neither	242	2.03 (1.23)	241	1.42 (1.08)
Public Safety Only	62	2.33 (1.30)	62	1.70 (1.13)
CAF Only	29	2.09 (1.27)	29	1.55 (1.14)
CAF and Public Safety	8	2.89 (1.02)	8	1.89 (0.85)
Test Statistic ^1^	-	*F* (3337) = 1.98	-	*F* (3336) = 1.48
Effect Size ($\eta_{p}^{2}$)	-	0.017	-	0.013

Notes. -: *n* = 0; ^: Sample size between 1 and 4, so data not presented. CAF = Canadian Armed Forces; CCG = Canadian Coast Guard; C&P = Conservation and Protection; MCTS = Marine Communications and Traffic Services; NWT = Northwest Territories; NVT = Nunavut; SD = Standard Deviation; YK = Yukon. ^1^ The test results comparing scores on organizational and operational stress measures across categorical participant demographics. Lettered superscripts within each column category indicate significant differences between category groups with different letters on outcome at *p* ≤ 0.05. ** *p* < 0.01—Statistically significantly different. Holm–Bonferroni adjustment applied to alpha levels to control Type I errors. Post hoc tests were not performed for some of the significant tests because at least one group had fewer than two cases.

**Table 3 ijerph-19-16396-t003:** Average Stress Levels Associated with Occupational Stressors.

	Total Sample	CCG	C&P	Comparing CCG and C&P	PSP-Total Sample	Comparing Total Sample to PSP-Total Sample
	*Mean (SD)*	*Mean (SD)*	*Mean (SD)*	Effect Size (Cohen’s d)	*Mean (SD)*	Effect Size (Cohen’s d)
Organizational Stressors						
Mean Organizational Stress Score	2.11 (1.25)	2.01 (1.24)	2.36 (1.26)	0.278*	3.62 (1.33)	1.208 ***
Dealing with co-workers	2.81 (1.87)	2.77 (1.87)	2.89 (1.86)	0.062	4.05 (1.78)	0.666 ***
The feeling that different rules apply to different people (e.g., favoritism)	2.57 (1.87)	2.55 (2.16)	2.63 (2.14)	0.038	4.15 (1.95)	0.735 ***
Feeling like you always have to prove yourself to the organization	2.67 (2.00)	2.61 (2.06)	2.79 (1.84)	0.091	4.02 (1.96)	0.677 ***
Excessive administrative duties	3.02 (2.02)	2.77 (2.00)	3.66 (1.93)	0.452 ***	3.69 (1.98)	0.331 ***
Constant change in policy/legislation	2.73 (2.00)	2.58 (2.01)	3.06 (1.93)	0.241 *	3.80 (1.91)	0.535 ***
Staff shortages	3.10 (2.17)	2.97 (2.00)	3.44 (2.00)	0.217	4.46 (2.08)	0.626 ***
Bureaucratic red tape	3.14 (2.01)	2.97 (2.00)	3.53 (2.00)	0.275 *	4.44 (1.98)	0.646 ***
Too much computer work	2.43 (1.94)	2.10 (1.84)	3.24 (1.97)	0.606 ***	3.19 (1.89)	0.391 ***
Lack of training on new equipment	2.00 (1.86)	2.11 (1.86)	1.74 (1.83)	0.199	3.12 (1.80)	0.603 ***
Perceived pressure to volunteer free time	1.20 (1.70)	1.22 (1.78)	1.13 (1.51)	0.053	2.64 (1.85)	0.845 ***
Dealing with supervisors	2.06 (2.01)	2.01 (2.00)	2.18 (2.04)	0.085	3.69 (2.00)	0.813 ***
Inconsistent leadership style	2.52 (2.20)	2.44 (2.20)	2.74 (2.19)	0.139	4.44 (2.10)	0.876 ***
Lack of resources	2.52 (2.17)	2.36 (2.17)	2.89 (2.14)	0.245*	4.29 (2.04)	0.816 ***
Unequal sharing of work responsibilities	2.30 (2.12)	2.21 (2.11)	2.54 (2.12)	0.155	3.90 (2.10)	0.755 ***
If you are sick or injured, your co-workers seem to look down on you	1.03 (1.65)	1.08 (1.62)	0.87 (1.65)	0.128	2.78 (2.00)	1.058 ***
Leaders over-emphasize the negatives (e.g., supervisor evaluations, public complaints)	1.37 (1.81)	1.28 (1.79)	1.58 (1.87)	0.163	3.51 (2.10)	1.183 ***
Internal investigations	0.92 (1.55)	.87 (1.51)	1.07 (1.64)	0.131	3.04 (2.08)	1.366 ***
Dealing with the court system	0.62 (1.30)	0.14 (0.60)	1.78 (1.74)	1.54 ***	2.63 (1.84)	1.550 ***
The need to be accountable for doing your job	1.64 (1.73)	1.59 (1.79)	1.77 (1.59)	0.100	3.35 (1.92)	0.986 ***
Inadequate equipment	1.71 (1.87)	1.70 (1.87)	1.76 (1.87)	0.029	3.13 (1.89)	0.759 ***
Operational Stressors						
Mean Operational Stress Score	1.49 (1.09)	1.39 (1.06)	1.73 (1.13)	0.318 **	3.17 (1.28)	1.539 ***
Shift work	1.31 (1.93)	1.26 (1.94)	1.42 (1.90)	0.088	3.32 (2.10)	1.039 ***
Working alone at night	0.64 (1.45)	0.64 (1.52)	0.63 (1.27)	0.010	2.21 (1.86)	1.080 ***
Overtime demands	1.43 (1.91)	1.35 (1.91)	1.65 (1.92)	0.157	2.68 (1.94)	0.654 ***
Risk of being injured on the job	1.09 (1.57)	1.00 (1.57)	1.31 (1.53)	0.194	2.98 (1.89)	1.214 ***
Work-related activities on days off (e.g., court, community events)	0.63 (1.30)	0.54 (1.28)	0.85 (1.34)	0.235 *	2.53 (1.77)	1.461 ***
Potentially psychologically traumatic events (e.g., motor vehicle accidents, domestics, death, injury)	0.89 (1.50)	0.85 (1.54)	1.02 (1.41)	0.117	3.39 (1.98)	1.664 ***
Managing your social life outside of work	1.61 (1.68)	1.54 (1.64)	1.76 (1.64)	0.131	3.02 (1.77)	0.858 ***
Not enough time available to spend with friends and family	2.10 (1.97)	2.01 (1.93)	2.27 (2.04)	0.135	3.54 (1.87)	0.732 ***
Paperwork	2.12 (1.79)	1.80 (1.68)	2.92 (1.82)	0.649 ***	3.29 (1.88)	0.651 ***
Eating healthy at work	1.88 (1.77)	1.88 (1.79)	1.88 (1.72)	0.002	3.40 (1.84)	0.862 ***
Finding time to stay in good physical condition	2.64 (1.97)	2.59 (1.99)	2.76 (1.96)	0.087	3.96 (1.85)	0.670 ***
Fatigue (e.g., shift work, overtime)	2.23 (2.08)	2.25 (2.12)	2.16 (1.97)	0.043	4.14 (1.99)	0.920 ***
Occupation-related health issues (e.g., back pain)	1.63 (1.81)	1.53 (1.75)	1.87 (1.95)	0.189	3.62 (2.02)	1.099 ***
Lack of understanding from family and friends about your work	1.34 (1.65)	1.24 (1.59)	1.60 (1.77)	0.219	3.04 (1.89)	1.031 ***
Making friends outside the job	1.65 (1.81)	1.61 (1.82)	1.78 (1.79)	0.095	2.73 (1.84)	0.597 ***
Upholding a “higher image” in public	1.36 (1.61)	1.21 (1.57)	1.73 (1.65)	0.327 **	2.98 (1.89)	1.012 ***
Negative comments from the public	1.37 (1.73)	1.12 (1.60)	1.99 (1.89)	0.516 ***	3.45 (1.97)	1.200 ***
Limitations to your social life (e.g., who your friends are, where you socialize)	1.37 (1.78)	1.21 (1.74)	1.76 (1.84)	0.309 *	2.99 (1.87)	0.910 ***
Feeling like you are always on the job	1.82 (2.01)	1.70 (2.00)	2.10 (2.02)	0.198	3.31 (2.01)	0.740 ***
Friends/family feel the effects of the stigma associated with your job	0.84 (1.47)	0.66 (1.36)	1.26 (1.65)	0.415 **	2.87 (1.88)	1.383 ***

Notes. * *p* < 0.05, ** *p* < 0.01, *** *p* < 0.001—Statistically significantly different. CAF = Canadian Armed Forces; CCG = Canadian Coast Guard; C&P = Conservation and Protection; PSP = Public Safety Personnel; SD = Standard Deviation.

**Table 4 ijerph-19-16396-t004:** Association Between Occupational Stressors and Positive Screens for Mental Health Disorders.

	PTSD	MDD	GAD	SAD	PD	AUD
	AOR (95% CI)	AOR (95% CI)	AOR (95% CI)	AOR (95% CI)	AOR (95% CI)	AOR (95% CI)
Organizational Stressors						
Mean Organizational Stress Score	1.70 (1.26, 2.30) ***	1.39 (1.09, 1.76) **	1.40 (1.05, 1.86) *	1.17 (0.90, 1.52)	1.77 (1.18, 2.66) **	2.21 (1.43, 3.42) ***
Dealing with co-workers	1.09 (0.90, 1.32)	0.97 (0.82, 1.13)	1.05 (0.88, 1.27)	1.11 (0.94, 1.32)	1.24 (0.96, 1.62)	1.56 (1.20, 2.04) ***
The feeling that different rules apply to different people (e.g., favoritism)	1.26 (1.06, 1.51) **	1.09 (0.95, 1.26)	1.06 (0.90, 1.24)	1.13 (0.97, 1.31)	1.28 (1.00, 1.62) *	1.36 (1.06, 1.76) *
Feeling like you always have to prove yourself to the organization	1.33 (1.10, 1.59) **	1.15 (0.99, 1.33)	1.12 (0.94, 1.33)	1.26 (1.07, 1.47) **	1.31 (1.03, 1.67) *	1.39 (1.08, 1.76) *
Excessive administrative duties	1.25 (1.04, 1.51) *	1.11 (0.96, 1.29)	1.11 (0.93, 1.32)	0.94 (0.81, 1.11)	1.15 (0.90, 1.49)	1.36 (1.06, 1.74) *
Constant change in policy/legislation	1.27 (1.05, 1.54) *	1.08 (0.94, 1.25)	1.02 (0.86, 1.21)	0.92 (0.78, 1.08)	1.29 (0.99, 1.68)	1.29 (1.01, 1.65) *
Staff shortages	1.08 (0.91, 1.28)	0.97 (0.84, 1.11)	1.13 (0.96, 1.34)	0.92 (0.80, 1.07)	1.35 (1.05, 1.73) *	1.23 (0.97, 1.55)
Bureaucratic red tape	1.32 (1.08, 1.62) **	1.13 (0.98, 1.31)	0.99 (0.83, 1.18)	0.94 (0.80, 1.11)	1.02 (0.78, 1.33)	1.34 (1.04, 1.72) *
Too much computer work	1.15 (0.95, 1.40)	1.19 (1.02, 1.38) *	1.22 (1.02, 1.46) *	1.04 (0.88, 1.23)	1.07 (0.83, 1.37)	1.44 (1.12, 1.86) **
Lack of training on new equipment	1.06 (0.88, 1.28)	1.26 (1.08, 1.46) **	1.21 (1.01, 1.44) *	1.18 (0.99, 1.39)	1.23 (0.96, 1.57)	1.31 (1.01, 1.70)
Perceived pressure to volunteer free time	1.21 (1.02, 1.45) *	1.24 (1.06, 1.45) **	1.31 (1.10, 1.56) **	1.19 (0.99, 1.41)	1.23 (0.98, 1.53)	1.26 (0.97, 1.62)
Dealing with supervisors	1.21 (1.03, 1.42) *	1.16 (1.01, 1.34) *	0.99 (0.84, 1.17)	1.04 (0.89, 1.21)	1.09 (0.88, 1.35)	1.39 (1.11, 1.75) **
Inconsistent leadership style	1.27 (1.08, 1.49) **	1.19 (1.04, 1.36) **	1.11 (0.95, 1.29)	1.03 (0.89, 1.19)	1.30 (1.04, 1.61) *	1.24 (1.00, 1.54) *
Lack of resources	1.21 (1.03, 1.41) *	1.12 (0.98, 1.28)	1.19 (1.01, 1.39) *	1.03 (0.90, 1.19)	1.34 (1.07, 1.66) **	1.22 (0.97, 1.52)
Unequal sharing of work responsibilities	1.26 (1.06, 1.48) **	1.24 (1.08, 1.43) **	1.21 (1.03, 1.42) *	1.04 (0.89, 1.20)	1.38 (1.10, 1.75) **	1.23 (0.99, 1.54)
If you are sick or injured, your co-workers seem to look down on you	1.57 (1.29, 1.90) ***	1.34 (1.14, 1.58) ***	1.32 (1.09, 1.58) **	1.27 (1.07, 1.53) **	1.54 (1.19, 1.98) ***	1.32 (1.02, 1.73) *
Leaders over-emphasize the negatives (e.g., supervisor evaluations, public complaints)	1.33 (1.11, 1.59) **	-	-	-	-	-
Internal investigations	1.15 (0.93, 1.41)	1.10 (0.92, 1.32)	1.16 (0.94, 1.44)	0.91 (0.73, 1.14)	1.21 (0.92, 1.59)	1.47 (1.12, 1.94) **
Dealing with the court system	0.90 (0.64, 1.28)	0.98 (0.75, 1.27)	1.23 (0.89, 1.71)	0.95 (0.69, 1.32)	1.53 (0.94, 2.47)	1.42 (0.91, 2.20)
The need to be accountable for doing your job	1.23 (1.02, 1.48) *	1.25 (1.07, 1.46) **	1.39 (1.16, 1.67) ***	1.26 (1.07, 1.50) **	1.39 (1.09, 1.76) **	1.39 (1.09, 1.79) **
Inadequate equipment	1.43 (1.19, 1.72) ***	1.10 (0.94, 1.29)	1.12 (0.94, 1.34)	1.12 (0.94, 1.32)	1.14 (.90, 1.44)	1.29 (1.01, 1.65) *
Operational Stressors						
Mean Operational Stress Score	2.16 (1.56, 3.00) ***	1.67 (1.28, 2.17) ***	1.77 (1.30, 2.40) ***	1.65 (1.23, 2.22) ***	2.19 (1.44, 3.33) ***	1.86 (1.21, 2.85) **
Shift work	1.20 (1.02, 1.42) *	0.97 (0.83, 1.13)	0.99 (0.83, 1.17)	1.09 (0.94, 1.27)	1.14 (0.92, 1.41)	1.27 (1.02, 1.57) *
Working alone at night	1.27 (1.02, 1.57) *	0.94 (0.76, 1.16)	1.08 (0.87, 1.34)	0.96 (0.78, 1.19)	1.21 (0.93, 1.57)	1.37 (1.07, 1.76) *
Overtime demands	1.24 (1.05, 1.46) *	1.16 (1.01, 1.34) *	1.34 (1.13, 1.58) ***	1.10 (0.94, 1.28)	1.26 (1.01, 1.55) *	1.35 (1.09, 1.67) **
Risk of being injured on the job	1.27 (1.04, 1.56) *	1.06 (0.89, 1.26)	1.22 (1.01, 1.48) *	1.05 (0.87, 1.28)	1.27 (0.98, 1.64)	1.66 (1.26, 2.18) ***
Work-related activities on days off (e.g., court, community events)	1.37 (1.08, 1.73) **	1.31 (1.06, 1.63) *	1.24 (0.97, 1.58)	1.25 (0.98, 1.60)	1.12 (0.80, 1.57)	0.87 (0.57, 1.34)
Potentially psychologically traumatic events (e.g., motor vehicle accidents, domestics, death, injury)	1.34 (1.09, 1.65) **	1.21 (1.01, 1.45) *	1.32 (1.07, 1.63) **	1.21 (0.97, 1.49)	1.53 (1.18, 1.99) ***	1.41 (1.05, 1.88) *
Managing your social life outside of work	1.57 (1.28, 1.93) ***	1.34 (1.13, 1.59) ***	1.44 (1.18, 1.75) ***	1.47 (1.21, 1.79) ***	1.69 (1.31, 2.20) ***	1.08 (0.82, 1.43)
Not enough time available to spend with friends and family	1.33 (1.13, 1.58) ***	1.26 (1.09, 1.44) ***	1.20 (1.02, 1.42) *	1.20 (1.03, 1.39) *	1.31 (1.05, 1.63) *	1.10 (0.87, 1.39)
Paperwork	1.25 (1.02, 1.53) *	1.29 (1.09, 1.53) **	1.27 (1.04, 1.54) *	1.08 (0.90, 1.29)	1.24 (0.94, 1.62)	1.10 (0.85, 1.44)
Eating healthy at work	1.30 (1.06, 1.58) **	1.14 (0.98, 1.34)	1.03 (0.86, 1.24)	1.04 (0.87, 1.23)	1.21 (0.94, 1.55)	1.27 (0.99, 1.63)
Finding time to stay in good physical condition	1.40 (1.15, 1.71) ***	1.27 (1.10, 1.48) **	1.22 (1.03, 1.45) *	1.11 (0.95, 1.31)	1.25 (0.99, 1.59)	1.27 (0.99, 1.62)
Fatigue (e.g., shift work, overtime)	1.26 (1.07, 1.49) **	1.19 (1.04, 1.36) **	1.19 (1.01, 1.39) *	1.11 (0.96, 1.28)	1.26 (1.01, 1.55) *	1.32 (1.06, 1.65) *
Occupation-related health issues (e.g., back pain)	1.28 (1.07, 1.53) **	1.14 (0.98, 1.33)	1.17 (0.99, 1.39)	1.09 (0.92, 1.28)	1.27 (1.01, 1.60) *	1.51 (1.17, 1.95) ***
Lack of understanding from family and friends about your work	1.26 (1.04, 1.52) *	1.24 (1.06, 1.46) **	1.32 (1.10, 1.60) **	1.29 (1.08, 1.54) **	1.25 (0.98, 1.59)	1.48 (1.14, 1.92) **
Making friends outside the job	1.55 (1.28, 1.88) ***	1.34 (1.14, 1.57) ***	1.19 (0.99, 1.43)	1.49 (1.25, 1.78) ***	1.41 (1.10, 1.81) **	0.94 (0.72, 1.22)
Upholding a “higher image” in public	1.34 (1.10, 1.63) **	1.26 (1.06, 1.48) **	1.28 (1.05, 1.56) *	1.43 (1.18, 1.74) ***	1.47 (1.13, 1.92) **	1.14 (0.88, 1.47)
Negative comments from the public	1.54 (1.26, 1.86) ***	1.27 (1.08, 1.49) **	1.29 (1.07, 1.55) **	1.32 (1.11, 1.58) **	1.79 (1.37, 2.34) ***	1.14 (0.89, 1.47)
Limitations to your social life (e.g., who your friends are, where you socialize)	1.45 (1.21, 1.74) ***	1.29 (1.10, 1.51) **	1.38 (1.15, 1.65) ***	1.42 (1.18, 1.69) ***	1.55 (1.22, 1.95) ***	1.11 (0.86, 1.44)
Feeling like you are always on the job	1.26 (1.07, 1.49) **	1.32 (1.14, 1.51) ***	1.31 (1.12, 1.54) ***	1.23 (1.05, 1.43) **	1.35 (1.09, 1.68) **	1.26 (1.02, 1.57) *
Friends/family feel the effects of the stigma associated with your job	1.51 (1.22, 1.88) ***	1.41 (1.17, 1.71) ***	1.43 (1.15, 1.76) ***	1.20 (0.97, 1.47)	1.34 (1.02, 1.75) *	1.51 (1.13, 2.01) **

Notes. AOR = Odds ratio adjusted for sex, age, education, marital status, province of residence, previous work experience, total number of PPTE types, and occupation category; AUD = Alcohol Use Disorder; CAF = Canadian Armed Forces; CCG = Canadian Coast Guard; CI = Confidence Interval; C&P = Conservation and Protection; GAD = Generalized Anxiety Disorder; MCTS = Marine Communications and Traffic Services; MDD = Major Depressive Disorder PD = Panic Disorder; PTSD = post-traumatic stress disorder; SAD = Social Anxiety Disorder. * *p* < 0.05, ** *p* < 0.01, *** *p* < 0.001—Statistically significantly different.

**Table 5 ijerph-19-16396-t005:** Association Between Potentially Psychologically Traumatic Event Exposures, Organizational Stressors, Operational Stressors, and Positive Screens for Mental Health Disorders.

	PTSD	MDD	GAD	SAD	PD	AUD
	AOR (95% CI)	AOR (95% CI)	AOR (95% CI)	AOR (95% CI)	AOR (95% CI)	AOR (95% CI)
Model 1						
Total Number of Potentially Psychologically Traumatic Event Exposure Types	1.13 (1.03, 1.25) **	1.05 (0.97, 1.12)	1.05 (0.96, 1.14)	1.02 (0.94, 1.10)	1.14 (1.00, 1.29) *	0.95 (0.85, 1.07)
Model 2						
Mean Organizational Stress Score	1.78 (1.34, 2.38) ***	1.39 (1.11, 1.76) **	1.42 (1.08, 1.86) *	1.17 (0.91, 1.50)	1.92 (1.30, 2.82) ***	1.89 (1.26, 2.86) **
Model 3						
Mean Operational Stress Score	2.25 (1.64, 3.09) ***	1.66 (1.29, 2.15) ***	1.76 (1.31, 2.36) ***	1.60 (1.20, 2.13) ***	2.37 (1.58, 3.55) ***	1.60 (1.08, 2.38) *
Model 4						
Total Number of Potentially Psychologically Traumatic Event Exposure Types	1.06 (0.95, 1.18)	1.00 (0.92, 1.08)	1.00 (0.91, 1.10)	0.98 (0.90, 1.07)	1.12 (0.96, 1.31)	0.87 (0.76, 0.99) *
Mean Organizational Stress Score	0.94 (0.59, 1.51)	0.80 (0.67, 1.37)	0.87 (0.56, 1.35)	0.66 (0.44, 1.01)	1.02 (0.56, 1.88)	2.31 (1.18, 4.54) *
Mean Operational Stress Score	2.27 (1.38, 3.74) ***	1.73 (1.17, 2.56) **	1.97 (1.23, 3.15) **	2.34 (1.46, 3.75) ***	2.15 (1.17, 3.97) *	0.94 (0.47, 1.87)
Model 5						
Trauma Exposure by Organizational Stress Interaction Term	1.03 (0.96, 1.11)	1.02 (0.97, 1.08)	1.03 (0.97, 1.10)	1.04 (0.98, 1.11)	1.01 (0.92, 1.12)	0.96 (0.87, 1.05)
Model 6						
Trauma Exposure by Operational StressInteraction Term	1.02 (0.94, 1.11)	0.99 (0.93, 1.06)	1.02 (0.95, 1.09)	1.04 (0.97, 1.11)	0.99 (0.89, 1.14)	0.99 (0.90, 1.09)

Notes. AOR = Odds ratio adjusted for sex, age, education, marital status, province of residence, previous work experience, and occupation category; AUD = Alcohol Use Disorder; CAF = Canadian Armed Forces; CCG = Canadian Coast Guard; CI = Confidence Interval; C&P = Conservation and Protection; GAD = Generalized Anxiety Disorder; MCTS = Marine Communications and Traffic Services; MDD = Major Depressive Disorder PD = Panic Disorder; PTSD = post-traumatic stress disorder; SAD = Social Anxiety Disorder. Model 1: The total number of potentially psychologically traumatic event types entered and adjusted for covariates. Model 2: The mean organizational stress score entered and adjusted for covariates. Model 3: The mean operational stress score entered and adjusted for covariates. Model 4: The total number of potentially psychologically traumatic event types, the mean organizational stress score, and the mean operational stress score entered into the same model simultaneously and adjusted for covariates. Model 5: Model 4 with the main effects of the total number of potentially psychologically traumatic event types, the mean organizational stress score, and the interaction term for the total number of potentially psychologically traumatic event types × the mean organizational stress score and adjusted for covariates and the mean operational stress score. Model 6: Model 4 with the main effects of the total number of potentially psychologically traumatic event types and the mean operational stress score and the interaction term for the total number of potentially psychologically traumatic event types x the mean operational stress score and adjusted for covariates and the mean organizational stress score. * *p* < 0.05, ** *p* < 0.01, *** *p* < 0.001—Statistically significantly different.

## Data Availability

Not applicable.

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
