# Peer review of "Assessing the Relative Impact of Diverse Stressors among Canadian Coast Guard and Conservation and Protection Officers"

_ijerph, 2022, doi:10.3390/ijerph192416396_

Round 1

Reviewer 1 Report

Dear authors, thank you for the opportunity to get acquainted with the results of your research.

The study is relevant and performed at a high level.

The introduction provides a detailed analysis of modern scientific research on the stated problem, substantiates the relevance. The introduction contains the objectives of the study and the author's assumptions.

Materials and methods describe the procedure for conducting the study, the sample of the study is described in detail and in detail. The wish to the authors is to better structure the description of research methods, as well as statistical methods, indicating all the factors taken into account and the application of all statistical procedures. It is possible to add information about which problem was solved with the help of which statistical methods.

The results of the study are described in detail and clearly illustrated by tables, the authors provide explanations for the tables.

The discussion of the results is extensive, correlating the results of this study with those of other authors, making generalizations, and listing in detail the limitations of the study. The authors need to add practical recommendations that can be given based on the results of the study, as well as possible directions for future research.

The conclusion reflects the main conclusions concisely.

There are some minor shortcomings in the design of the article, in particular, in the discussion of the results section (position of the text on the pages, landscape orientation of the text, its break).

In connection with the above, the article can be recommended for publication after the elimination of minor comments.

Regards and best wishes, the reviewer

Author Response

Dear authors, thank you for the opportunity to get acquainted with the results of your research.

The study is relevant and performed at a high level.

The introduction provides a detailed analysis of modern scientific research on the stated problem, substantiates the relevance. The introduction contains the objectives of the study and the author's assumptions.

Materials and methods describe the procedure for conducting the study, the sample of the study is described in detail and in detail. The wish to the authors is to better structure the description of research methods, as well as statistical methods, indicating all the factors taken into account and the application of all statistical procedures. It is possible to add information about which problem was solved with the help of which statistical methods.

Thank you for your positive comments and feedback to strengthen the manuscript. We appreciate your time and effort spent reviewing the manuscript and hope you will find our updates satisfactory and the manuscript suitable for publication. We have updated the study objectives, statistical methods, and results to be structured the same to allow for the objective, methods, and results to clarify and to allow the reader to connect each objective with the corresponding statistical method and results.

The results of the study are described in detail and clearly illustrated by tables, the authors provide explanations for the tables.

The discussion of the results is extensive, correlating the results of this study with those of other authors, making generalizations, and listing in detail the limitations of the study. The authors need to add practical recommendations that can be given based on the results of the study, as well as possible directions for future research.

Thank you for identifying this opportunity to strengthen the manuscript. We have updated the discussion to include practical recommendations and some directions for further research.

The conclusion reflects the main conclusions concisely.

There are some minor shortcomings in the design of the article, in particular, in the discussion of the results section (position of the text on the pages, landscape orientation of the text, its break).

Thank you for identifying this area to improve the manuscript. We have moved the tables out of the text and to the end of the results. This has removed the separation between the text in the results section and the changes in the page orientation.

In connection with the above, the article can be recommended for publication after the elimination of minor comments.

Regards and best wishes, the reviewer

Reviewer 2 Report

Comments for authors.

The article they present is very interesting, therefore it captures the attention of the readers; since it is a topic that covers the different factors of work stress.

In my opinion, only some changes would have to be made in the tables to make them more understandable; for example, in table 3 on page 21 you can see some values that are missing the zero before the point.

On the other hand, in the conclusions part, a little more could be covered about how the studies carried out contributed to having these results.

Author Response

The article they present is very interesting, therefore it captures the attention of the readers; since it is a topic that covers the different factors of work stress.

In my opinion, only some changes would have to be made in the tables to make them more understandable; for example, in table 3 on page 21 you can see some values that are missing the zero before the point.

Thank you for your positive comments and feedback to strengthen the manuscript. We appreciate your time and effort spent reviewing the manuscript and hope you will find our updates satisfactory and the manuscript suitable for publication.

Thank you for identifying this. We have updated the tables and addressed all the missing zero values.

On the other hand, in the conclusions part, a little more could be covered about how the studies carried out contributed to having these results.

Thank you for identifying this opportunity to strengthen the manuscript. We have updated the discussion to expand on the implications of the current results including practical recommendations and some directions for further research.